# High Prevalence of Smoking-Related Diseases in High-Grade and Muscle-Invasive Bladder Cancer: Opportunities for Lung Cancer Screening

**DOI:** 10.3390/cancers17233741

**Published:** 2025-11-23

**Authors:** Riccardo Regazzo, Federica Ciccarese, Simone Paglialonga, Elio Renò, Caterina Gaudiano, Beniamino Corcioni, Francesco Chessa, Riccardo Schiavina, Cristina Mosconi

**Affiliations:** 1Department of Radiology, IRCCS Azienda-Ospedaliero Universitaria di Bologna, 40138 Bologna, Italy; 2Department of Medical and Surgical Sciences (DIMEC), University of Bologna, 40136 Bologna, Italy; 3Division of Urology, IRCCS Azienda-Ospedaliero Universitaria di Bologna, 40138 Bologna, Italy

**Keywords:** smoking-related diseases, bladder cancer, lung cancer, lung cancer screening

## Abstract

Early detection is crucial for improving outcomes in many cancers. Screening programs aim to identify disease at an early, treatable stage, but to be effective, they must focus on people at highest risk. In lung cancer, newer risk models now consider not only smoking but also prior cancers and lung conditions. Because smoking is a major shared risk factor for both bladder cancer and many lung and heart diseases, our study investigated how often these conditions occur in patients with invasive bladder cancer. We found that many also had lung or heart disease detectable on CT scans. This suggests that chest CT, already part of bladder cancer staging, may provide added value by simultaneously detecting other clinically relevant conditions, including pulmonary nodules suspicious for early-stage lung cancer.

## 1. Introduction

Bladder carcinoma (BC) is the tenth most frequently diagnosed malignancy worldwide. Its prevalence varies across regions, reflecting differences in exposure to risk factors and access to diagnostic resources. Globally, the estimated incidence is 9.5 per 100,000 person-years in men and 2.4 per 100,000 person-years in women [1]. Urothelial carcinoma accounts for approximately 95% of cases, although other histological variants—including squamous cell carcinoma, adenocarcinoma, and the rarer neuroendocrine or leiomyomatous tumors—have been described [2,3]. Clinically, BC most commonly presents with painless macroscopic hematuria, although abdominal pain and lower urinary tract symptoms such as dysuria and urgency may also occur [4]. Ultrasound is often used as the initial imaging modality in patients with hematuria, but its sensitivity is limited for small or anatomically hidden tumors [5]. Urine cytology offers high specificity (>90%) in high-grade BC or carcinoma in situ (CIS), yet sensitivity remains low in detecting low-grade lesions [6]. Definitive diagnosis relies on cystoscopy and, when indicated, transurethral resection of the bladder tumor (TURB) for histopathological evaluation. Histopathological assessment includes tumor grading, according to the World Health Organization system, which classifies lesions as papillary urothelial neoplasm of low malignant potential, low-grade (LG), or high-grade (HG) carcinoma [7]. Staging follows the TNM classification, and the presence of detrusor muscle invasion critically determines both treatment strategies and radiological follow-up. Non–muscle-invasive BC (NMIBC; stages Ta, T1, and CIS) occurs in about 75% of cases and is generally managed conservatively with TURB, with or without subsequent intravesical immunotherapy or chemotherapy. In contrast, muscle-invasive BC (MIBC; stage T2 or higher) carries a worse prognosis and requires radical treatment with cystectomy, systemic chemotherapy, and, in selected cases, radiotherapy [8]. Detrusor invasion is not only the key prognostic factor but also determines the need for staging investigations. According to the European Association of Urology (EAU) guidelines, cross-sectional imaging with CT or MRI is not routinely indicated in patients with NMIBC or LGBC. Instead, staging CT of the chest, abdomen, and pelvis is specifically recommended in patients with MIBC or HGBC in order to evaluate lymph node involvement and distant metastases [8,9]. Both inherited and environmental factors contribute to BC development. Established risk factors include exposure to chemical agents such as aromatic amines or polycyclic aromatic hydrocarbons, pelvic radiotherapy, and chronic schistosomiasis. However, tobacco smoking remains the predominant acquired risk factor, accounting for a substantial proportion of BC cases worldwide [10,11,12]. Importantly, smoking is also strongly associated with other diseases, including atherosclerosis and pulmonary disorders such as pulmonary emphysema, interstitial lung diseases (ILDs) and lung cancer (LC) [13,14,15]. Despite these well-known associations, the prevalence of smoking-related diseases (SRDs) in patients with BC has not been systematically investigated. High-resolution computed tomography (HRCT) of the chest represents the diagnostic gold standard for SRDs, and low-dose HRCT has been shown to reduce LC mortality in large screening cohorts. These studies highlight the importance of selecting patients from high-risk populations, both to optimize screening efficacy and to minimize false positives and overdiagnosis, thereby improving the cost-effectiveness ratio of screening programs [16,17,18]. Given that BC and many SRDs share tobacco exposure as a common etiological factor, patients with BC may represent a population at particularly high risk for both prevalent SRDs and possibly for the development of primary LC. Investigating the prevalence of SRDs in this cohort may help clarify whether BC patients could represent an ideal target group for structured HRCT-based screening programs.

We retrospectively evaluated the prevalence of SRDs in a cohort of patients with histologically confirmed HGBC or MIBC. The risk of SRDs was compared among current smokers, former smokers, and non-smokers. As a secondary objective, we assessed the prevalence of pulmonary nodules suspicious for primary LC and compared the risk of LC across patient groups.

## 2. Materials and Methods

### 2.1. Population of the Study

This study was conducted at the Radiology Unit of Sant’Orsola University Hospital, Bologna, Italy. We included patients with MIBC OR HGBC who underwent staging CT between June 2021 and June 2025. The CT protocol comprised scans of the abdomen, pelvis, and chest, including HRCT, both before and after administration of intravenous contrast medium. Patients were excluded if chest CT was not included in the imaging protocol. The final study cohort consisted of 166 patients who had undergone at least one HRCT of the chest prior to cystectomy. When multiple CT scans were available, only the most recent examination was analyzed. Collected clinical data included sex, age, smoking status, histological grade and TNM stage of BC. Patients were categorized as current smokers, former smokers or non-smokers depending on their smoking habits. The presence of SRDs, including suspicious pulmonary nodules, pulmonary emphysema, airway disease, and ILD, was assessed for each patient. When available, histological data of suspicious pulmonary nodules, obtained either by bronchoscopic biopsy or surgical excision, were also collected. This study is an observational, retrospective single-center study and was approved by our local institution review board. Informed consent was waived by the institutional review board owing to the retrospective nature of the study.

### 2.2. CT Acquisition Technique

All staging CT acquisitions were performed with patients in the supine position during end-inspiration, both before and after the administration of intravenous contrast medium. Chest CT scans were performed on two different CT scanners:GE Medical System, LightSpeed VCT 64-slice CT: Technical parameters included a tube voltage of 120 kV, tube current modulation ranging from 100 to 250 mAs, a spiral pitch factor of 0.98, and a collimation width of 64 × 0.625 mm. Reconstructions were performed using a BONEPLUS convolution kernel at a slice thickness of 1.25 mm.PHILIPS Ingenuity CT 128-slice CT: Technical parameters included a tube voltage of 120 kV, tube current modulation ranging from 100 to 250 mAs, a spiral pitch factor of 1.224, and a collimation width of 64 × 0.625 mm. Reconstructions were performed using a Y-SHARP convolution kernel at a slice thickness of 1 mm.

Expiration scans were not performed.

### 2.3. Imaging Analysis

All CT scans were independently reviewed by two blinded radiologists with substantial expertise in chest imaging. Any disagreements in imaging findings or classifications were resolved by consensus between the two readers. Pulmonary nodules were considered suspicious for primary LC when no evidence of bladder carcinoma metastases or involvement of the common iliac lymph nodes (N3 stage) was present. In these cases, the radiological probability of pulmonary malignancy was further characterized according to the Lung-RADS criteria [19], and only nodules with a Lung-RADS score ≥3 were included in the estimation of LC prevalence. The severity of emphysema and airway disease was classified according to the Fleischner Society guidelines [20], with moderate to severe emphysema (grades 1C, 1D, 1E, or 3B) considered indicative of smoking-related disease. Coronary artery calcifications (CAC) were evaluated in accordance with the Society of Thoracic Radiology recommendations [21], and only moderate to severe CAC (grades 2 or 3) were included in the analysis. Interstitial abnormalities suggestive of ILD were also documented.

### 2.4. Statistical Analysis

Categorical variables were summarized as counts and percentages, while numerical variables were summarized as mean ± standard deviation, as well as minimum and maximum values. Binary logistic regression was employed to investigate associations between smoking status and each smoking-related disease. Smoking status was treated as an independent variable, while each disease was modeled as a binary outcome. Odds ratios (ORs) with 95% confidence intervals (CIs) were calculated. Statistical significance was set at *p* < 0.05. All analyses were performed using R (version 4.5.1). We performed a post-hoc analysis to evaluate the statistical precision of our estimate of the prevalence of suspicious pulmonary nodules (Lung-RADS ≥ 3) in our cohort of 166 patients. Using an exact binomial method, we calculated a 95% confidence interval (CI) for the observed detection rate of 6.6%, which ranged from 3.4% to 11.5%, resulting in an effective two-sided margin of error of approximately ±3.8%. This analysis provides an assessment of the variability and potential uncertainty associated with our prevalence estimate.

## 3. Results

Of the 166 patients with muscle invasive or high-grade bladder carcinoma, 37 (22%) were female and 129 (78%) were male, with a mean age of 72.9 ± 10.3 years. Eighty-nine patients (53.6%) were non-smokers, while 77 (46.4%) had a history of smoking (28 current smokers and 49 former smokers). Overall, 100 patients (60.2%) had at least one SRD as defined by imaging criteria, including pulmonary disease or moderate to severe coronary artery calcifications. SRD prevalence did not differ significantly between male and female patients. Current smokers had a significantly higher risk of SRD compared with non-smokers (*p* < 0.05). When current and former smokers were combined, SRD prevalence remained significantly higher than in non-smokers (*p* < 0.05). Former smokers alone showed a higher SRD prevalence compared with non-smokers (65.3% vs. 52.8%, OR 1.68), but this difference was not statistically significant. The prevalence of SRDs among smokers and non-smokers is summarized in Table 1 and visually represented in Figure 1.

Smoking-related pulmonary disease (*p*-SRD) was identified in 53 patients (31.9%), and moderate to severe CAC (grades 2 or 3, Society of Thoracic Radiology) was present in 86 (51.8%). Although CAC prevalence did not differ significantly between smokers and non-smokers, smokers had a significantly higher risk of *p*-SRD (*p* < 0.01). Moderate to severe centrilobular or paraseptal emphysema (grades 1C, 1D, 1E, or 3B, Fleischner Society) was significantly more frequent in smokers than in non-smokers (*p* < 0.01). Among smokers, current smokers had a higher risk of emphysema compared with former smokers (*p* < 0.01). An illustrative example of severe emphysema (grade 1D) diagnosed in one of our patients is shown in Figure 2.

Interstitial abnormalities suggestive of early interstitial lung disease (ILD) and airway pathology were found in 23 patients (13.9%) and 13 patients (7.8%), respectively, without significant differences between smokers and non-smokers. Nodules suspicious for LC (Lung-RADS score ≥ 3) were present in 11 patients (6.6%), each with a single lesion. Of these, 4 were current smokers, 4 were former smokers, and 3 were non-smokers. Although the prevalence of suspicious nodules was higher among patients with a history of smoking (72.7% vs. 27.3%, OR 3.32), the difference did not reach statistical significance. Histological data, obtained either via bronchoscopic biopsy or surgical excision, were available for five of these eleven patients (45%). Importantly, all nodules were confirmed as primary lung cancers (three adenocarcinomas and two squamous cell carcinomas), excluding metastatic bladder carcinoma. A representative example of a suspicious pulmonary nodule (Lung-RADS score = 4B) detected in one of the patients is shown in Figure 3.

## 4. Discussion

Our study demonstrated a high prevalence (60.2%) of SRDs in patients with HGBC or MIBC, a population that shares tobacco exposure as a major risk factor. Both current smokers and the overall group of smokers showed a significantly increased risk of developing SRDs. However, SRD prevalence did not differ significantly between former smokers and never-smokers. This finding may be explained by the beneficial effects of early and sustained smoking cessation on pulmonary function, as demonstrated in previous studies [22,23,24]. A considerable prevalence of *p*-SRDs (31.9%) was observed, with emphysema emerging as the most frequent pulmonary disease and showing the strongest association with smoking status. Importantly, current smokers were significantly more affected than former smokers, highlighting the protective role of smoking cessation in limiting emphysema development [25].

While smoking is a well-established risk factor for both pulmonary emphysema and coronary atherosclerosis [26], and a strong relationship between these conditions has been documented [27,28], our cohort did not show a significantly higher prevalence of moderate-to-severe CAC among smokers compared to non-smokers. This may be due to the characteristics of our “non-smoker” group, which consisted of relatively older patients (mean age 73 ± 11 years) likely to have accumulated other cardiovascular risk factors—such as hypertension, dyslipidemia, or diabetes—that substantially contribute to atherosclerosis, potentially acting as confounding variables that obscured a clearer association between smoking status and CAC prevalence in our cohort. Future studies controlling for such variables are necessary to elucidate the independent effects of smoking on coronary atherosclerosis and improve risk stratification strategies.

In our population, 6.6% of patients had a solitary pulmonary nodule (a Lung-RADS score of 3 or higher) that was suspicious for primary LC. None of these patients had evidence of metastatic bladder carcinoma, and based on radiological features, patient history, and staging, pulmonary nodules were reasonably attributed to primary LC rather than BC metastases. Notably, histological analysis confirmed a primary lung origin for all nodules where data were available, reinforcing the diagnostic reliability of Lung-RADS criteria. However, it is important to acknowledge that histological confirmation was available for only 45% of cases, and the absence of complete histological data precludes the calculation of a definitive false-positive rate for radiological findings alone in this specific cohort. Our observed prevalence (6.6%) is comparable to findings by O’Dwyer Et Al., who reported an LC prevalence of 6.4% in a screened population with a mean age of 66 years and a history of prior malignancies [29]. This supports the notion that pulmonary comorbidities and prior cancer history should be considered when developing integrated risk-prediction models for LC. Previous studies have already demonstrated that risk-prediction tools such as the PLCOM2012 model, which incorporates pulmonary comorbidities and prior malignancies, improve the efficiency of LC detection, and reduce false positives in screening programs [30,31].

Interestingly, 45% of patients with suspicious pulmonary nodules also presented with moderate-to-severe emphysema, reinforcing the value of incorporating comorbidities into LC screening eligibility criteria. These findings underscore the importance of carefully selecting high-risk patients for screening, both to reduce false positives and to optimize cost-effectiveness. To our knowledge, this is the first study to systematically investigate the prevalence of SRDs in patients with HGBC/MIBC. Our results support the inclusion of HRCT among staging imaging examinations for BC, as currently recommended by the EAU guidelines [8,9]. Beyond detecting pulmonary nodules suggestive of metastases, HRCT may also identify early-stage, treatable LC, pulmonary diseases, and signs of coronary atherosclerosis requiring further pneumological or cardiological assessment. Given the shared etiological link with smoking habits and the observed high prevalence of SRDs and suspicious pulmonary nodules in our cohort, expanding HRCT surveillance to NMIBC patients with significant smoking histories could enable earlier detection of treatable pulmonary disease and primary LC. Future prospective studies are needed to evaluate the clinical utility and cost-effectiveness of such an approach.

This study has several limitations that should be acknowledged. First, the relatively small sample size may have limited the precision of our prevalence estimates for suspicious pulmonary nodules and primary lung cancer. Although our post-hoc analysis suggests that these estimates are reasonably reliable, the limited number of histological confirmations—available for only 45% of detected nodules—prevents definitive conclusions regarding their malignancy status and the exact false-positive rate. The potential for false positives warrants caution, as unnecessary invasive procedures could be performed on benign lesions, underscoring the need for larger, prospective validation studies.

Second, as a single-center, retrospective analysis, our findings may be affected by selection biases and may not be generalizable to broader population settings. Demographic factors such as age and baseline risk profiles might differ from those in other cohorts, further limiting applicability. Additionally, due to the study’s retrospective nature, we were unable to systematically collect data on other important risk factors—including hypertension, diabetes, hyperlipidemia, and other comorbidities—that could influence findings related to coronary artery calcifications and the prevalence of SRDs. We also lacked detailed data on pack-years of smoking. This limitation restricts our capacity to control for potential confounders that might impact the observed associations.

Despite these constraints, our results highlight the role of chest HRCT in the comprehensive staging of high-risk bladder cancer patients, emphasizing the importance of further research. Future large-scale, multicenter prospective studies are necessary to validate these preliminary findings, refine prevalence estimates, and determine the true diagnostic and prognostic value of incorporating HRCT into routine screening protocols for patients with BC or other high-risk groups.

## 5. Conclusions

Patients with BC show a high prevalence of SRDs, particularly among current and former smokers, that could significantly impact patient prognosis and quality of life, and they particularly benefit from early detection and sustained smoking cessation. Our findings further support the utility of chest HRCT in detecting SRDs, and potentially primary early-stage lung cancer, during staging in patients with HGBC/MIBC. HRCT is currently recommended primarily for MIBC patients, who represent a minority of bladder cancer cases, while it is not routinely performed for patients with NMIBC. Given the shared etiological link between smoking and both BC and LC, coupled with the high prevalence of SRDs observed in our smoking population, we propose that HRTC could be considered for all patients with BC with a significant smoking history, including those with NMIBC.

Given the limitations discussed earlier, further large-scale, prospective studies are essential to thoroughly evaluate the prognostic value and cost-effectiveness of routine chest HRCT screening in current and former smokers.

## Figures and Tables

**Figure 1 cancers-17-03741-f001:**
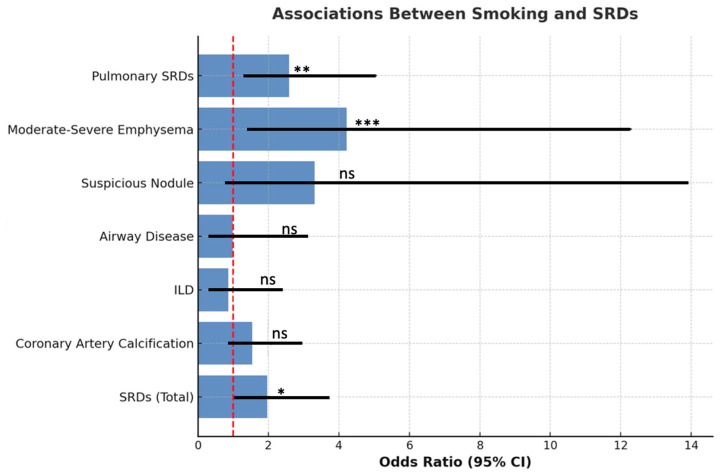
Odds ratios (ORs) with 95% confidence intervals (CIs) for smoking-related diseases (SRDs) in the study population. The red dashed line indicates the reference value (OR = 1). Asterisks represent statistical significance levels: *** *p* < 0.001; ** *p* < 0.01; * *p* < 0.05; ns = non-significant.

**Figure 2 cancers-17-03741-f002:**
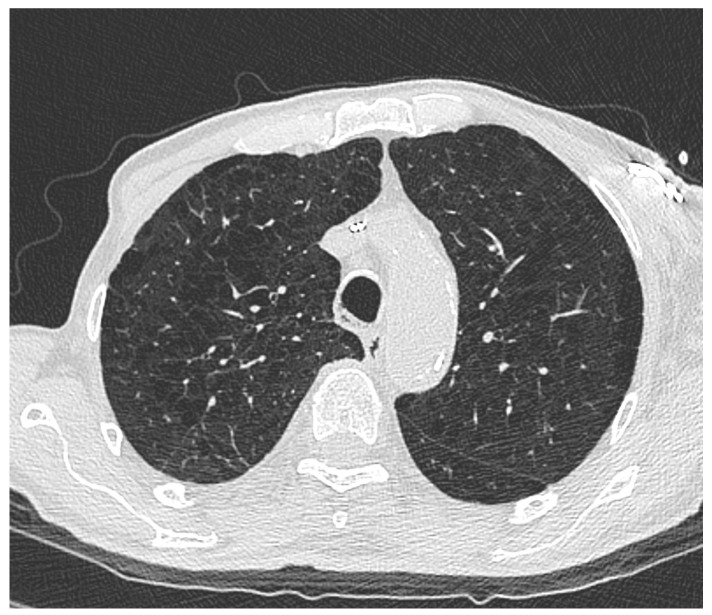
Severe centrilobular and paraseptal emphysema (grade 1D) in both upper pulmonary lobes in a 67-year-old current male smoker.

**Figure 3 cancers-17-03741-f003:**
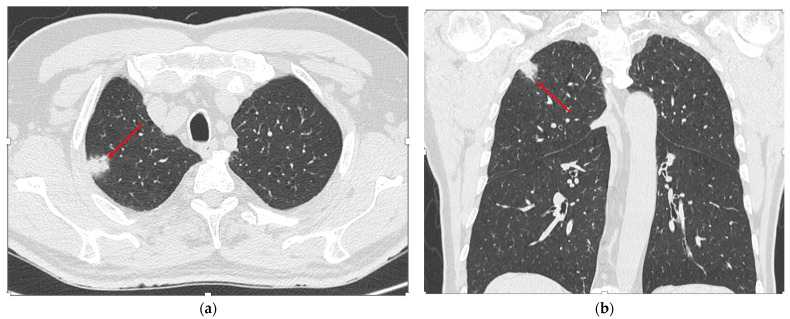
Axial (**a**) and coronal (**b**) HRCT images highlighting a 20 mm suspicious pulmonary nodule (Lung-RADS 4B) in the right upper lobe of a 75-year-old male smoker, with arrows pointing to key findings.

**Table 1 cancers-17-03741-t001:** Prevalence of smoking-related diseases (SRDs) in patients with histologically confirmed HGBC/MIBC.

	Current Smokers (n = 28)	Former Smokers (n = 49)	Non-Smokers (n = 89)	Smokers vs. Non-Smokers OR (95% CI), *p*-Value
**Pulmonary SRDs**	16	17	20	**2.59 (1.32–5.07), *p* < 0.01**
Moderate—severe emphysema	14	4	6	**4.22 (1.45–12.31), *p* < 0.01**
Suspicious nodule	4	4	3	3.32 (0.79–13.91), *p* = 0.97
Airway disease	3	3	7	0.99 (0.32–3.08), *p* = 0.99
ILD	4	6	13	0.87 (0.32–2.36), *p* = 0.77
**Coronary artery calcifications**	14	28	44	1.54 (0.88–2.68), *p* = 0.13
**SRDs (Total)**	21	32	47	**1.97 (1.04–3.74), *p* = 0.03**

Statistically significant results are indicated in bold.

## Data Availability

Data analyzed during the current study are available from the corresponding author upon reasonable request (data are not publicly available due to privacy restrictions).

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
