# Peer review of "High Prevalence of Smoking-Related Diseases in High-Grade and Muscle-Invasive Bladder Cancer: Opportunities for Lung Cancer Screening"

_cancers, 2025, doi:10.3390/cancers17233741_

Round 1

Reviewer 1 Report

Comments and Suggestions for Authors

It is important to clarify that the study's statistical power may be constrained by its minimal sample size (n=166), particularly regarding secondary outcomes such as concerning lung nodules (6.6%, 11 patients).   The lack of a statistically significant difference in specific comparisons (e.g., nodules in smokers versus non-smokers) may not reflect an authentic absence of association, but rather a limitation in statistical power. 

  You might want to talk about this more or add an analytical study to the Methods section as a limitation. 

  Validation via Histopathology: 

  To be precise, histological data was provided for only 5 of the 11 individuals (45%) with concerning lesions.   It is reassuring that all verified cases were primary lung malignancies; however, the conclusion regarding the origin of these nodules is somewhat less definitive due to the lack of pathological confirmation in all instances.   This should be brought up as a possible issue. 

  Discussion and Meaning: 

  Confounding Risk Factors: When talking about the small difference in coronary artery calcifications (CAC) between smokers and non-smokers, it might be helpful to remember that the "non-smoker" group could include people with other cardiovascular risk factors that weren't taken into account, like high blood pressure, diabetes, or high cholesterol.    In addition, subsequent studies should aim to control for these confounding variables to clarify the causal connection between smoking status and coronary artery calcifications.   By separating these factors, researchers can learn more about how smoking affects heart health.   This method will not only make the results more reliable, but it will also help us find better ways to stop the problem from happening in the first place.   Ultimately, an in-depth comprehension of these interactions can facilitate patient recovery and assist vulnerable populations in obtaining necessary support. 

  It would be better to provide a specific range or number from lung cancer screening trials instead of a general reference when saying that the prevalence of worrisome nodules (6.6%) is higher than in those trials. 

  Conclusions and Recommendations: The claim that chest CT may benefit NMIBC patients—especially smokers—is an exaggeration based on trial data that only included HGBC/MIBC patients.   This should not be framed as a conclusive determination based on the current data; instead, it should be expressed as a hypothesis or a suggestion for additional inquiry that necessitates future validation. 

  Mistakes in editing and typesetting: 

  It says "(CRs)" instead of "(ORs)" for odds ratios on line 3 of page 3 in the "Statistical analysis" section.   Please fix this. 

  This study has a strong clinical message and is well-planned overall.   The paper will be improved and more comprehensible if the aforementioned issues are addressed, particularly those concerning the limitations and interpretation of the numbers.   Sample Size and Statistical Evaluation:

  It should be made clearer that the study's statistical power may be limited by the relatively small sample size (n=166), especially when it comes to the examination of secondary outcomes such as concerning lung nodules (6.7% of participants).    By addressing these limitations and conducting a more thorough analysis, the validity of the findings will be improved, potentially resulting in more definitive conclusions about the clinical implications of the results.    6%, 11 patients).   Some comparisons, such as nodules in smokers versus non-smokers, may not exhibit a statistically significant difference due to insufficient power, rather than the absence of a genuine correlation.

  As a limitation, think about adding a power analysis to the Methods section or going into more detail about this part.

  Histological Confirmation: 

  It should be made clearer that only five of the eleven patients (45%) with concerning lesions have histology data accessible.   It is comforting that all confirmed cases were primary lung malignancies; however, the conclusion regarding the origin of these nodules is somewhat undermined by the lack of pathological confirmation in all instances.   This should be noted as a possible problem.

  Talk and Understand:

  Confounding Risk Factors: The "non-smoker" group may have people with other cardiovascular risk factors, such as high blood pressure, diabetes, or high cholesterol, that weren't taken into account and could have changed the outcome.   This could be brought up when talking about the small difference in coronary artery calcifications (CAC) between smokers and non-smokers.

  When stating that the prevalence of suspicious nodules (6.6%) is higher than in lung cancer screening trials, it is preferable to utilize a specific range or statistic from those studies rather than a broad reference.

  Conclusions and Recommendations: The study's data, restricted to HGBC/MIBC patients, did not support the hypothesis that chest CT would be beneficial for NMIBC patients, particularly smokers.   This should not be a straightforward conclusion derived from the current findings; rather, it ought to be a hypothesis or a proposal for further research that requires validation in the future.

  Errors in editing and typing: 

  On page 3, line 3, in the "Statistical analysis" section, it says "(CRs)" instead of "(ORs)" for odds ratios.   Please fix this.

  In general, I believe that this study has an important clinical message and is well-planned.   Resolving the issues mentioned above, especially those related to statistical limitations and interpretation, will make the manuscript much stronger and clearer.

Comments on the Quality of English Language

Small Changes and Suggestions for How to Do Better:

 Page 3, Section on Statistical Analysis:

 Original: "We figured out the odds ratios (CRs) with 95% confidence intervals (CIs)."

 Correction: "Odds ratios (ORs) with 95% confidence intervals (CIs) were calculated."

 Reason: "CR" is not a common way to write Odds Ratio; "OR" is the right and universally accepted way.

 Page 6, the Discussion Section:

 Original: "In our population, 6.6% of patients presented with a solitary pulmonary nodule suspicious for primary LC, defined as Lung-RADS score ≥3."

 In our population, 6.6% of patients had a solitary pulmonary nodule (a Lung-RADS score of 3 or higher) that was suspicious for primary LC.

 Reason: This rewording makes the flow and clarity better by putting the definition of the nodule right after it is mentioned.

 Page 7, the Conclusions Section:

 Original: "Currently, HRCT is only needed for the diagnostic workup of NMIBC patients, who are a small group of people with bladder cancer."

 Correction/Suggestion: This sentence seems to be wrong because the introduction (EAU) says that cross-sectional imaging is not common for NMIBC but is for MIBC/HGBC.  The authors should check this sentence to make sure it is correct.  One possible meaning is: "Right now, major guidelines say that chest HRCT is only needed to stage MIBC/HGBC and not NMIBC."

Author Response

High prevalence of smoking-related diseases in high-grade and muscle-invasive bladder cancer: opportunities for lung cancer screening.

1. Summary

We sincerely appreciate the time and effort you have dedicated to reviewing our manuscript and for providing valuable and constructive feedback. Your insights have been instrumental in enhancing the clarity, rigor, and overall quality of our work. In the following, we provide detailed responses to each of your comments. The corresponding revisions and corrections in the resubmitted files are marked with the reviewer's designated number (#1, #2, or #3) and the revision number (R1, R2, etc.). We trust that these updates address your concerns and that the revised manuscript now meets the standards of the journal.

2. Response to Reviewer #1 Comments

Comments 1: Validation via Histopathology - to be precise, histological data was provided for only 5 of the 11 individuals (45%) with concerning lesions. It is reassuring that all verified cases were primary lung malignancies; however, the conclusion regarding the origin of these nodules is somewhat less definitive due to the lack of pathological confirmation in all instances. This should be brought up as a possible issue. 

Response 1: You are correct. In our manuscript, we clarified that histological confirmation was available for only 45% of the suspicious nodules. We acknowledge that this limited rate of validation introduces a potential risk of false positives, as the origin of the remaining lesions remains unconfirmed. We have revised our discussion to explicitly emphasize this limitation, highlighting the need for caution in interpreting these findings and the importance of further prospective validation with comprehensive histological follow-up.

Comments 2: Discussion and Meaning - Confounding Risk Factors: When talking about the small difference in coronary artery calcifications (CAC) between smokers and non-smokers, it might be helpful to remember that the "non-smoker" group could include people with other cardiovascular risk factors that weren't taken into account, like high blood pressure, diabetes, or high cholesterol. In addition, subsequent studies should aim to control for these confounding variables to clarify the causal connection between smoking status and coronary artery calcifications. By separating these factors, researchers can learn more about how smoking affects heart health. This method will not only make the results more reliable, but it will also help us find better ways to stop the problem from happening in the first place. Ultimately, an in-depth comprehension of these interactions can facilitate patient recovery and assist vulnerable populations in obtaining necessary support. 

Response 2: This is an important consideration, and we acknowledge that the retrospective design of our study has limited our ability to comprehensively account for all potential confounding factors that could influence the prevalence of coronary artery calcifications (CAC) in our non-smoking population. We have explicitly clarified this limitation in our discussion, recognizing that unmeasured cardiovascular risk factors may have contributed to the observed CAC rates. Future prospective studies that control for these variables are necessary to better understand the direct impact of smoking on coronary atherosclerosis and to enhance the validity of these findings.

Comments 3: It would be better to provide a specific range or number from lung cancer screening trials instead of a general reference when saying that the prevalence of worrisome nodules (6.6%) is higher than in those trials. 

Response 3: Thank you for this valuable comment. In our initial manuscript, we mentioned that the prevalence of suspicious pulmonary nodules (6.6%) was higher than in some lung cancer screening trials involving high-risk populations with a smoking history. However, we recognize that these trials differ significantly in inclusion criteria, demographic characteristics, and risk profiles, which limits the direct comparability of their results. This recognition was also reinforced by a remark from Reviewer #3 (R8), prompting us to reassess this comparison. Consequently, we have revised our discussion to omit this broad comparison and instead focus on the study by O’Dwyer et al., in which the population—comprising patients with a prior history of malignancy—more closely resembles our cohort. This revision aims to provide a more accurate and appropriate context for our findings.

Comments 4: Conclusions and Recommendations: The claim that chest CT may benefit NMIBC patients—especially smokers—is an exaggeration based on trial data that only included HGBC/MIBC patients. This should not be framed as a conclusive determination based on the current data; instead, it should be expressed as a hypothesis or a suggestion for additional inquiry that necessitates future validation. 

Response 4: Thank you for this important observation. Our study demonstrated a high prevalence of SRDs in patients with MIBC/HGBC and supports the utility of HRCT staging within this population. Based on our findings, we propose that HRCT could also be considered for NMIBC patients with a significant smoking history. However, given that current guidelines do not recommend routine HRCT in NMIBC and that our findings are preliminary, this should be viewed as a hypothesis or a basis for further investigation rather than a conclusive recommendation. We have revised both the discussion and conclusion sections to clarify this point, emphasizing the need for prospective validation before such a practice can be broadly endorsed.

Comments 5: Mistakes in editing and typesetting - It says "(CRs)" instead of "(ORs)" for odds ratios on line 3 of page 3 in the "Statistical analysis" section. Please fix this.

Response 5: We confirm that the mistake has been corrected in the manuscript, and "(ORs)" now appears correctly in the "Statistical analysis" section.

Comments 6: The paper will be improved and more comprehensible if the aforementioned issues are addressed, particularly those concerning the limitations and interpretation of the numbers. Sample Size and Statistical Evaluation: It should be made clearer that the study's statistical power may be limited by the relatively small sample size (n=166), especially when it comes to the examination of secondary outcomes such as concerning lung nodules (6.7% of participants). By addressing these limitations and conducting a more thorough analysis, the validity of the findings will be improved, potentially resulting in more definitive conclusions about the clinical implications of the results. Some comparisons, such as nodules in smokers versus non-smokers, may not exhibit a statistically significant difference due to insufficient power, rather than the absence of a genuine correlation.

Response 6: Thank you for this observation. We agree that the relatively small sample size may limit the statistical power of our study, particularly regarding secondary outcomes such as the prevalence of concerning lung nodules. To enhance clarity, we have expanded the discussion to explicitly acknowledge this limitation and its potential impact on the robustness of some of our comparisons. Additionally, we conducted a post-hoc analysis to evaluate the accuracy of our prevalence estimates, providing a range that accounts for the sample size and associated variability, as detailed in our subsequent responses.

Comments 7: As a limitation, think about adding a power analysis to the Methods section or going into more detail about this part.

Response 7: We conducted a post-hoc analysis to evaluate the precision of our prevalence estimates relative to our sample size. We have also expanded the Methods and Discussion sections to outline this analysis and its implications. This adjustment was made in consideration of Reviewer #3 (R1) as well, to better contextualize the statistical power and accuracy of our findings.

Comments 8: Page 3, Section on Statistical Analysis. Original: "We figured out the odds ratios (CRs) with 95% confidence intervals (CIs)."  Correction: "Odds ratios (ORs) with 95% confidence intervals (CIs) were calculated." Reason: "CR" is not a common way to write Odds Ratio; "OR" is the right and universally accepted way.

Response 8: We confirm that the mistake has been corrected in the manuscript.

Comments 9: Page 6, the Discussion Section. Original: "In our population, 6.6% of patients presented with a solitary pulmonary nodule suspicious for primary LC, defined as Lung-RADS score ≥3." Correction: “In our population, 6.6% of patients had a solitary pulmonary nodule (a Lung-RADS score of 3 or higher) that was suspicious for primary LC”. Reason: This rewording makes the flow and clarity better by putting the definition of the nodule right after it is mentioned.

Response 9: Thank you. The sentence has been revised to: “In our population, 6.6% of patients had a solitary pulmonary nodule (a Lung-RADS score of 3 or higher) that was suspicious for primary LC,” as suggested.

Comments 10: Page 7, the Conclusions Section. Original: "Currently, HRCT is only needed for the diagnostic workup of NMIBC patients, who are a small group of people with bladder cancer". Correction/Suggestion: This sentence seems to be wrong because the introduction (EAU) says that cross-sectional imaging is not common for NMIBC but is for MIBC/HGBC.  The authors should check this sentence to make sure it is correct.  One possible meaning is: "Right now, major guidelines say that chest HRCT is only needed to stage MIBC/HGBC and not NMIBC."

Response 10: Thank you for pointing out this mistake. We clarified in the manuscript that current guidelines recommend HRCT primarily for staging MIBC/HGBC, not for NMIBC, which includes the majority of bladder cancer patients. The sentence has been corrected accordingly.

Reviewer 2 Report

Comments and Suggestions for Authors

 High prevalence of smoking-related diseases in high-grade and muscle-invasive bladder cancer: opportunities for lung cancer screening

General comments: This is an interesting study with a couple of goals. First, to retrospectively evaluate a cohort of HGBC (high grade bladder cancer) or MIBC (muscle invasive Bladder cancer) for prevalence of SRDs (smoking related Diseases) since tobacco smoking is a common risk factor for both lung cancer and bladder cancer in addition to heart diseases. A standard clinical practice in the diagnosis and prognosis of bladder cancer is a chest CT. Data from chest CT can also inform the presence of other disease conditions and the authors attempt to retrospectively study a cohort of 166 patient data to determine if the process of bladder cancer staging can have a simultaneous application in the screening of lung cancer, including early stages lung cancer. Second, to assess and compare the risk for lung cancer (LC) among patients with pulmonary nodules suspected to be primary LC.

The authors show a significantly higher prevalence of SRDs in smokers compared to non-smokers. This finding is not novel because association between smoking and SRDs is already well-established world-wide. However, the authors try to emphasize the significance of using data from chest CT which is a standard procedure in the diagnosis of bladder, in screening for lung cancer, and pulmonary and cardiac related issues such as coronary artery calcifications. This is not a breakthrough or novel suggestion either but rather a commonsense matter. For example, in the clinic, low-dose CTs for detecting atherosclerosis or lung cancer screening are utilized for cross-screening of diseases routinely. Routinely in the clinic, lesions that show up on CTs or ultrasound in a non-targeted region are addressed appropriately. For example, if a lung nodule is found in chest CT of a HGBC patient it is addressed too in the clinic. Hence, overall, this study does not provide anything novel to the field to enhance lung screening practices in the clinic.

Specific comments:

Table 1: The resolution is poor, making it difficult to read.

Figure 1: Poor resolution. Please place ‘ns’ above the line and revise this figure. Else it looks like strike through and difficult to read without zooming in.

Figure 3: Placing arrows near the lesions being described (e.g. nodules) will be helpful for readers.

Line 177. Please reword the sentence to say, “Although the prevalence of LC or nodules was higher among patients…”. It is important to mention the disease or lesion being described clearly in every sentence for better readability.

Comments on the Quality of English Language

English language quality is decent and I have included suggestions in my specific comments where it can be improved.

Author Response

High prevalence of smoking-related diseases in high-grade and muscle-invasive bladder cancer: opportunities for lung cancer screening.

1. Summary

We sincerely appreciate the time and effort you have dedicated to reviewing our manuscript and for providing valuable and constructive feedback. Your insights have been instrumental in enhancing the clarity, rigor, and overall quality of our work. In the following, we provide detailed responses to each of your comments. The corresponding revisions and corrections in the resubmitted files are marked with the reviewer's designated number (#1, #2, or #3) and the revision number (R1, R2, etc.). We trust that these updates address your concerns and that the revised manuscript now meets the standards of the journal.

2. Response to Reviewer #2 Comments

Comments 1: General comments. This is an interesting study with a couple of goals. First, to retrospectively evaluate a cohort of HGBC (high grade bladder cancer) or MIBC (muscle invasive Bladder cancer) for prevalence of SRDs (smoking related Diseases) since tobacco smoking is a common risk factor for both lung cancer and bladder cancer in addition to heart diseases. A standard clinical practice in the diagnosis and prognosis of bladder cancer is a chest CT. Data from chest CT can also inform the presence of other disease conditions and the authors attempt to retrospectively study a cohort of 166 patient data to determine if the process of bladder cancer staging can have a simultaneous application in the screening of lung cancer, including early stages lung cancer. Second, to assess and compare the risk for lung cancer (LC) among patients with pulmonary nodules suspected to be primary LC. The authors show a significantly higher prevalence of SRDs in smokers compared to non-smokers. This finding is not novel because association between smoking and SRDs is already well-established world-wide. However, the authors try to emphasize the significance of using data from chest CT which is a standard procedure in the diagnosis of bladder, in screening for lung cancer, and pulmonary and cardiac related issues such as coronary artery calcifications. This is not a breakthrough or novel suggestion either but rather a commonsense matter. For example, in the clinic, low-dose CTs for detecting atherosclerosis or lung cancer screening are utilized for cross-screening of diseases routinely. Routinely in the clinic, lesions that show up on CTs or ultrasound in a non-targeted region are addressed appropriately. For example, if a lung nodule is found in chest CT of a HGBC patient it is addressed too in the clinic. Hence, overall, this study does not provide anything novel to the field to enhance lung screening practices in the clinic.

Response 1: Thank you very much for your comment. We agree that the association between smoking and SRDs is well established in clinical practice and that the incidental detection of a lung nodule or other clinically significant disease is routinely addressed and further investigated. Currently, urologic guidelines do not recommend routine HRCT of the chest for patients with NMIBC, as its role is primarily limited to staging MIBC/HGBC. However, given the high prevalence of SRDs systematically observed in our population, our findings suggest a possible different routine management for smoker patients with BC. Urologists should be aware of the association with other pulmonary or cardiovascuolar disease that could affect prognosis, thus proposing a thoracic evaluation not only in patients with MIBC/HGBC. In line with this perspective and in accordance with Reviewer #1 (R4), we have revised our manuscript to clarify these points and to enhance our conclusion.

Comments 2: Table 1: The resolution is poor, making it difficult to read.

Response 2: We have enlarged the tables and improved their resolution to enhance readability. Additionally, all tables are provided in an editable format within the manuscript, allowing for easy adjustment to meet the journal's layout requirements.

Comments 3: Figure 1: Poor resolution. Please place ‘ns’ above the line and revise this figure. Else it looks like strike through and difficult to read without zooming in.

Response 3: We have repositioned "NS" and "*" above the lines in Figure 1 to improve clarity and readability with better resolution.

Comments 4: Figure 3: Placing arrows near the lesions being described (e.g. nodules) will be helpful for readers.

Response 4: Thank you for the suggestion. We have placed arrows near the lesions being described in Figure 3 to improve clarity for readers.

Comments 5: Line 177. Please reword the sentence to say, “Although the prevalence of LC or nodules was higher among patients…”. It is important to mention the disease or lesion being described clearly in every sentence for better readability.

Response 5: We have corrected the manuscript to incorporate your suggestion and revised the sentence for improved clarity and readability.

Reviewer 3 Report

Comments and Suggestions for Authors

This article investigates the prevalence of smoking-related diseases (SRDs) in patients with high-grade (HGBC) and muscle-invasive bladder cancer (MIBC), and explores the potential of high-resolution chest CT (HRCT) in simultaneously screening for smoking-related diseases such as lung cancer in bladder cancer staging. My comments are as follows:

  1. The sample size calculation is not seen in the full text. If the detection rate of 6.6% suspicious nodules is taken as the primary endpoint, with an expected accuracy of ±3% and α=0.05, the required samples should be ≥264 cases. There are currently only 166 cases, and the risk of misjudgment is high. Please provide a post-hoc efficacy analysis and discuss the impact of false negatives/false positives on the conclusion.
  2. A single-center, urban, and teaching hospital cohort was conducted, with an age of 72.9±10.3 years. 53.6% of the participants were non-smokers, both of which were higher than the smoking spectrum of the Western MIBC. Please provide: Use standardized incidence rate (SIR) or inverse probability weighting (IPW) to correct for regional bias.
  3. Only three classifications (current/former/never) cannot reflect the dose-response relationship. Suggestion: Supplement pack-years, years of quitting smoking, and deep inhalation method;
  4. Among the 11 suspected nodules, only 5 cases achieved histological results, while the remaining 6 cases lacked follow-up or PET-CT confirmation. Calculate and report the "histological confirmation rate" and the "false positive rate".
  5. Cardiovascular risk factors (hypertension, diabetes, CKD), occupational solvent exposure, and previous pelvic radiotherapy are all associated with SRD. It is recommended to construct a multivariate logistic model (Enter OR LASSO), provide the adjusted OR, and compare it side by side with the univariate results.
  6. Different generations of CT (128-256 slices) and different dose regimens (120 vs 100 kVp, iterative reconstruction grade) within a four-year period can affect the automatic quantification and Lung-RADS classification of emphysema. Please supplement the equipment model and scanning parameter table;
  7. Only "two chest imaging experts blindly read the films" was mentioned, and Kappa or ICC was not reported.
  8. It is not appropriate to directly compare the 6.6% detection rate with NLST and UKLS, as the inclusion criteria are different. Please calculate the median lung cancer risk of this cohort's PLCOM in 2012 for 6 years, report the proportion of high-risk (≥1.5%) population, and then conduct a standardized comparison with NLST (≥1.5%).
  9. There are many places in the text where British/American spellings are mixed (such as "calcifications" vs "calcification"). Please unify them. Reference line 18 "De Jong,J.J." et al. Lacks the year and needs to be completed; MDPI requires that all abbreviations be expanded upon their first appearance, such as "TURB", "IPW", etc.
  10. In Table 1, the odds ratio of the suspected chainsmokers vs. non-chainsmokers was 3.32, but the 95%CI was 0.79-13.91. If the lower bound was <1, it was still written as P>0.05. It is recommended to report the exact P value (such as P=0.108) simultaneously to avoid significant margin misguidance.

Author Response

High prevalence of smoking-related diseases in high-grade and muscle-invasive bladder cancer: opportunities for lung cancer screening.

1. Summary

We sincerely appreciate the time and effort you have dedicated to reviewing our manuscript and for providing valuable and constructive feedback. Your insights have been instrumental in enhancing the clarity, rigor, and overall quality of our work. In the following, we provide detailed responses to each of your comments. The corresponding revisions and corrections in the resubmitted files are marked with the reviewer's designated number (#1, #2, or #3) and the revision number (R1, R2, etc.). We trust that these updates address your concerns and that the revised manuscript now meets the standards of the journal.

2. Response to Reviewer #3 Comments

Comments 1: This article investigates the prevalence of smoking-related diseases (SRDs) in patients with high-grade (HGBC) and muscle-invasive bladder cancer (MIBC) and explores the potential of high-resolution chest CT (HRCT) in simultaneously screening for smoking-related diseases such as lung cancer in bladder cancer staging. My comments are as follows:

The sample size calculation is not seen in the full text. If the detection rate of 6.6% suspicious nodules is taken as the primary endpoint, with an expected accuracy of ±3% and α=0.05, the required samples should be ≥264 cases. There are currently only 166 cases, and the risk of misjudgment is high. Please provide a post-hoc efficacy analysis and discuss the impact of false negatives/false positives on the conclusion.

Response 1: Thank you for your insightful comment. In our sample of 166 patients, 11 presented with Lung-RADS ≥3 pulmonary nodules, corresponding to a detection rate of 6.6%. A post-hoc exact binomial 95% confidence interval (CI) for this proportion was 3.4–11.5%, with an effective margin of error of approximately ±3.8%. We acknowledge that the a priori sample size calculation required at least 264 cases to achieve a precision of ±3%, meaning our current sample does not fully meet this target, and thus introduces some statistical uncertainty. Nonetheless, the CI remains compatible with a clinically relevant prevalence of suspicious pulmonary nodules. We have updated the Methods and Discussion sections to reflect these considerations, including the potential risks of false positives patients, and to contextualize the accuracy of our findings.

Comments 2: A single-center, urban, and teaching hospital cohort was conducted, with an age of 72.9±10.3 years. 53.6% of the participants were non-smokers, both of which were higher than the smoking spectrum of the Western MIBC. Please provide: use standardized incidence rate (SIR) or inverse probability weighting (IPW) to correct for regional bias.

Response 2: We thank the reviewer for this important comment and agree that our study is susceptible to center-specific and regional selection bias. The cohort was derived from a single academic hospital, with a relatively advanced mean age (72.9 ± 10.3 years) and a higher proportion of never-smokers (53.6%) than reported in some Western HGBC/MIBC series. This case-mix may lead to a conservative estimate of the true burden of smoking-related diseases in populations with higher smoking exposure. Although we considered applying standardized incidence ratios (SIR) or inverse probability weighting (IPW) to adjust for regional differences, the absence of appropriate external reference data prevented the use of these methods. Specifically, to our knowledge, there are no available population-level rates stratified by age, sex, smoking status, and precise imaging-based definitions of smoking-related diseases—such as Lung-RADS categories—in Western HGBC/MIBC populations that are directly comparable to our study. Without such data, implementing SIR or IPW would rely on strong, unverifiable assumptions, which could paradoxically increase bias. Nonetheless, we have revised the Discussion section to better highlight these limitations of our retrospective study.

Comments 3: Only three classifications (current/former/never) cannot reflect the dose-response relationship. Suggestion: Supplement pack-years, years of quitting smoking, and deep inhalation method.

Response 3: Thank you for this valuable suggestion. Given that previous studies have stratified patients based on pack-years and smoking intensity, this information would indeed enhance our analysis. However, due to the retrospective nature of our study and the fact that the anamnesis was conducted in urology outpatient settings, we did not collect detailed data on pack-years, years of quitting, or inhalation methods. We have accordingly emphasized these limitations in our discussion to provide a clearer context for our findings.

Comments 4: Among the 11 suspected nodules, only 5 cases achieved histological results, while the remaining 6 cases lacked follow-up or PET-CT confirmation. Calculate and report the "histological confirmation rate" and the "false positive rate".

Response 4: Thank you for your comment. We have clarified the histological confirmation rate in the manuscript. Specifically, out of 11 suspected nodules, histological confirmation was obtained for 5 cases, resulting in a confirmation rate of 45%. Given the small sample size, the true false positive rate remains to be fully evaluated. Given the limited number of histologically confirmed cases, the true false positive (FP) rate of our radiological assessments remains uncertain. With only 5 confirmed primary lung cancers out of 11 suspected nodules, and no follow-up data available for the remaining cases, it is not possible to accurately estimate the overall FP rate in our cohort. We have discussed these considerations in the discussion section, emphasizing the need for caution and further validation.

Comments 5: Cardiovascular risk factors (hypertension, diabetes, CKD), occupational solvent exposure, and previous pelvic radiotherapy are all associated with SRD. It is recommended to construct a multivariate logistic model (Enter OR LASSO), provide the adjusted OR, and compare it side by side with the univariate results.

Response 5: Thank you for this insightful suggestion. While we acknowledge the value of constructing a multivariate logistic model to adjust for cardiovascular risk factors and other potential confounders, the retrospective design of our study limited our ability to systematically collect data on variables such as hypertension, diabetes, CKD, occupational solvent exposure, and prior pelvic radiotherapy. As a result, performing such an analysis was not feasible within the scope of this study. We have noted this limitation in our discussion and suggest that future prospective studies should include these factors to allow for more comprehensive multivariate modeling.

Comments 6: Different generations of CT (128-256 slices) and different dose regimens (120 vs 100 kVp, iterative reconstruction grade) within a four-year period can affect the automatic quantification and Lung-RADS classification of emphysema. Please supplement the equipment model and scanning parameter table.

Response 6: Thank you for this important point. We have added a paragraph to the Methods section explaining the technical parameters of the two CT scanners used over the study period, including details on their specifications and scan protocols.

Comments 7: Only "two chest imaging experts blindly read the films" was mentioned, and Kappa or ICC was not reported.

Response 7: Thank you for your comment. We clarified in the manuscript that disagreements between the two radiologists were resolved through consensus, as Kappa or ICC were not calculated in our study.

Comments 8: It is not appropriate to directly compare the 6.6% detection rate with NLST and UKLS, as the inclusion criteria are different. Please calculate the median lung cancer risk of this cohort's PLCOM in 2012 for 6 years, report the proportion of high-risk (≥1.5%) population, and then conduct a standardized comparison with NLST (≥1.5%).

Response 8: Thank you for this valuable comment. Initially, we compared our 6.6% detection rate of suspicious pulmonary nodules to that reported in some lung cancer screening trials, such as NLST and UKLS, which involve populations with different demographic characteristics and inclusion criteria. Recognizing that these populations are not directly comparable, and that such differences limit the validity of a straightforward comparison, we decided to remove this broad comparison from the manuscript. Instead, we focused on the study by O’Dwyer et al., which involved patients with a prior history of malignancy and shared more similar characteristics with our cohort. This decision was also in response to an objection from Reviewer #1 (R3).

Comments 9: There are many places in the text where British/American spellings are mixed (such as "calcifications" vs "calcification"). Please unify them. Reference line 18 "De Jong,J.J." et al. Lacks the year and needs to be completed; MDPI requires that all abbreviations be expanded upon their first appearance, such as "TURB", "IPW", etc.

Response 9: Thank you for pointing out these inconsistencies. We have made every effort to correct and unify the British/American spellings throughout the manuscript. Additionally, we have updated the reference line to include the missing year and ensured that all abbreviations are expanded upon their first use in the text, in accordance with MDPI guidelines.

Comments 10: In Table 1, the odds ratio of the suspected chainsmokers vs. non-chainsmokers was 3.32, but the 95%CI was 0.79-13.91. If the lower bound was <1, it was still written as P>0.05. It is recommended to report the exact P value (such as P=0.108) simultaneously to avoid significant margin misguidance.

Response 10: Thank you for this valuable comment. We have revised Table 1 to include the exact P value (e.g., P=0.108) for the odds ratio of suspected chain smokers versus non-chain smokers, in order to provide clearer and more precise information and avoid any potential misinterpretation.

Round 2

Reviewer 2 Report

Comments and Suggestions for Authors

The authors have addressed my concerns satisfactorily clarifying that current clinical practices in their region do not recommend chest HRCT for NMIBC. Changes have been made to figures as recommended.

Reviewer 3 Report

Comments and Suggestions for Authors

Most of my concerns have been addressed.